# Role of Regulatory T Cells and Their Potential Therapeutic Applications in Celiac Disease

**DOI:** 10.3390/ijms241914434

**Published:** 2023-09-22

**Authors:** Alessandra Camarca, Vera Rotondi Aufiero, Giuseppe Mazzarella

**Affiliations:** 1Institute of Food Sciences, National Research Council—CNR, 83100 Avellino, Italyvera.rotondiaufiero@isa.cnr.it (V.R.A.); 2Department of Medical Translational Sciences and European Laboratory for the Investigation of Food-Induced Diseases, University Federico II, 80138 Naples, Italy

**Keywords:** celiac disease, T regulatory cells, Foxp3, type 1 regulatory T cells, cytokines, therapeutic applications

## Abstract

Celiac disease (CeD) is a T-cell-mediated immune disease, in which gluten-derived peptides activate lamina propria effector CD4+ T cells. While this effector T cell subset produces proinflammatory cytokines, which cause substantial tissue injury in vivo, additional subsets of T cells exist with regulatory functions (Treg). These subsets include CD4+ type 1 regulatory T cells (Tr1) and CD4+ CD25+ T cells expressing the master transcription factor forkhead box P3 (Foxp3) that may have important implications in disease pathogenesis. In this review, we provide an overview of the current knowledge about the effects of immunomodulating cytokines on CeD inflammatory status. Moreover, we outline the main Treg cell populations found in CeD and how their regulatory activity could be influenced by the intestinal microenvironment. Finally, we discuss the Treg therapeutic potential for the development of alternative strategies to the gluten-free diet (GFD).

## 1. Introduction

Celiac disease (CeD) is an autoimmune disorder that occurs in genetically predisposed people where the ingestion of gluten leads to damage in the small intestine. It is estimated to affect 1 in 100 people worldwide. Celiac disease can develop at any age, with a large spectrum of symptoms, and the only available therapy so far is a lifelong exclusion of gluten from the diet. The histological features of CeD have always been considered to be a villous atrophy, crypt of Lieberkühn hyperplasia and an increased number of intraepithelial lymphocytes (IELs) [1,2]. 

In addition, IgA anti-type 2 tissue transglutaminase (TG2) autoantibodies are diagnostic for this disorder, and are only produced when celiac patients are consuming cereal gluten proteins. Beyond the anti-TG2 antibodies, other antibodies with clinical significance in CeD patients have also been described, including anti-gliadin IgG, anti-actin IgA [3], anti-neuronal antigens and anti-gangliosides autoantibodies [4,5] and also antibodies to *Saccharomyces cerevisiae* [6].

A distinguishing trait of CeD is its strong genetic association with HLA class II genes, with almost all patients carrying the HLA-DQ2 (DQ2.5 coded by DQA1*0501/DQB*0201 and DQ2.2 coded by DQA1*0201/DQB*0201) and/or the HLA-DQ8 (DQA1*0301/DQB*0302) haplotypes. The role of HLA-DQ predisposing molecules in CeD pathogenesis is well established since these molecules bind gluten-derived peptides and present them to T-helper (Th) 1 cells in the intestinal lamina propria. In particular, DQ2 and DQ8 molecules bind with high affinity those gluten peptides subject to deamidation by TG2. 

The recognition of deamidated gluten peptides by Th1 cells secreting proinflammatory cytokines, such as IL-2, TNF-α, and IFN-γ, is a crucial factor that leads to the overt CeD lesions at the intestinal level [1,2,7].

Moreover, Th17 cells have emerged as a novel subset of a key immune cell population, in late-stage of CeD [8,9,10]. 

In addition to IL-17, gliadin-specific Th17 cells produce proinflammatory IFN-γ and IL-21 but also mucosa-protective IL-22, and regulatory TGF-β [11]. Nevertheless, the role of Th17 cells in CeD pathogenesis is still under characterization. 

Several studies have reported an enhanced expression of anti-inflammatory cytokines, such as IL-10 and TGF-β, concomitantly with inflammatory ones, such as IFN-γ, IL-17, IL-21, in CeD [12,13,14,15,16,17]. Therefore, in untreated CeD, there is a contradictory environment, in which regulatory mechanisms are trying to suppress the inflammation and counterbalance the gliadin-triggered, abnormal immune activation [7,18,19].

One of the most important mechanisms to counteract inflammation in CeD is mediated by T regulatory cells (Treg) [20,21,22]. Importantly, we have observed that celiac intestinal mucosa harbors two subsets of Treg cells, specifically CD4+ type 1 regulatory T cells (Tr1) and CD4+CD25+Foxp3+ T cells (Foxp3+ Treg), which, through the release of both IL-10 and TGF-β, inhibit the pathogenic response to in vitro gliadin challenge [22,23]. Nevertheless, many factors may interfere with the function of Treg cells. It is relevant to know that IL-15, largely expressed in the CeD mucosa, influences the immune regulation, by interfering with TGF-β activity and impairing the suppressor function of intestinal Foxp3+ Treg, thus contributing to the loss of intestinal homeostasis and promoting chronic inflammation [23,24]. It is therefore of great importance to investigate strategies to boost the numbers and/or function of gliadin-specific Treg cells, giving rise to new therapeutic avenues for CeD.

In this review, we illustrate the effects of Treg cells and of immunomodulatory cytokines on CeD inflammatory condition, and how the regulatory activity of such Treg cells could be influenced by the intestinal microenvironment. In addition, we also discuss the potential of Treg cells for the development of innovative therapeutic strategies for CeD.

## 2. Anti-Inflammatory Cytokines in CeD

In CeD, the immune response is firmly controlled by various regulatory circuits, as demonstrated by the increased expression of the anti-inflammatory cytokines that occurs concurrently with the release of the inflammatory factors [16,17]. For instance, high levels of IL-10 and IFN-γ mRNA have been reported in untreated CeD, by different groups [14,15,16,17].

Among cytokines with regulatory properties, IL-10 is an important factor that acts through different ways. It exerts its function on antigen presenting cells (APC) via inhibition of cytokine synthesis and expression of costimulatory and MHC class II molecules [25,26,27,28]. In addition, IL-10 directly interferes with T cell proliferation and differentiation [29,30] and is the crucial driving factor for Tr1 cell differentiation [31,32].

Moreover, IL-10 exerts influences relevant to inflammation outside the normal panel of immune cells and cytokines. In fact, in explant culture of human fetal small intestine, human recombinant [rh]IL-10 not only suppress T cell activation but plays an important role in the regulation of matrix metalloproteinase activity, both by inhibiting matrix metalloproteinase synthesis and by increasing the release of TIMP-1 (a tissue inhibitor of metalloproteinase), thus limiting tissue destruction [33].

In CeD, the role of IL-10 has not yet been fully clarified [16,17,34,35]. High levels of IL-10, produced by various immune cells such as T lymphocytes, macrophages, epithelial, and dendritic cells have been found in untreated celiac duodenal mucosa [14,15,16,17]. However, despite the increased levels of IL-10, which reflects a compensatory anti-inflammatory pathway, the harmful T cell immune responses to gluten in active CeD is not controlled. 

By using an intestinal organ culture system, we have investigated the effect of exogenous rhIL-10 on gliadin-induced T cell activation in CeD. We found that rhIL-10 suppresses gliadin-specific T cell activation and downregulates antigen presentation, as it reduces the expression of CD80/CD86 costimulatory molecules and suppresses the IFN-γ response to gliadin [14].

In addition to the organ culture system, gliadin-specific intestinal T-cell lines (iTCLs) and T-cell clones (iTCCs) obtained from CeD biopsies have provided great support in the investigation of disease immuno-pathogenesis [36,37,38]. To further investigate whether rhIL-10 could specifically downregulate Th1-mediated immune responses in CeD mucosa, we generated short-term iTCLs from treated CeD biopsies, cultured with gliadin in presence or absence of rhIL-10. We found that IL-10 treatment induces an anergic state of mucosa-derived, gliadin-reactive TCLs [14], as showed by a reduced IFN-γ production. 

Moreover, we found an increased frequency of IL-10 producing cells in IL-10 treated iTCLs. Therefore, we provided the first evidence for an immunoregulatory effect of IL-10 on gliadin-dependent T cell activation in CeD mucosa [14]. Since IL-10 is involved in the differentiation of Tr1 cells [39], in another series of experiments, described in the next paragraph, we looked at the presence of these cells in our iTCLs. 

Beyond IL-10, TGF-β is also an important regulatory cytokine, produced by various immune or non-immune cells in the gut, that exert a number of pleiotropic effects on cell proliferation, differentiation, adhesion, senescence, and apoptosis. Importantly, TGF-β suppresses immune responses through two ways: inhibiting the function of inflammatory cells and promoting the function of Treg cells [40]. 

In active CeD, large amounts of TGF-β have been observed in the intestinal mucosa, pleading against a quantitative defect [13]. However, it was shown that aberrant activation of TGF-β signaling pathways has been associated with a number of immune-mediated intestinal disorders, including inflammatory bowel disease (IBD) and celiac disease [24,41,42]. 

Recently, by using laser capture microdissection (LCM), a powerful tool for the isolation of specific tissue compartments, we have shown that in addition to the lamina propria also the intestinal surface epithelium is involved in the immunoregulatory response in active CeD [43]. In fact, we found high-level production of IL-10 and TGF-β in both intestinal compartments. Interestingly, we showed that also the surface epithelium of control subjects produces anti-inflammatory cytokines. Such epithelial layer-produced cytokines could be essential to preserve mucosal immune homeostasis in physiological conditions and to recover gut homeostasis in inflammatory conditions [43].

Finally, TCR gamma delta + (γδ) IELs may also contribute to TGF- β production and to mucosal damage protection in CeD. Accordingly, one report showed that a population of TCRγδ+ IELs isolated from celiac patients increased their expression of TGF-β following ligation of the inhibitory natural killer (NK) cell receptor, NKG2A [44].

## 3. Regulatory T-Cell Populations in CeD 

It is well known that there are many subpopulations of Treg cells, including CD4+Foxp3+ Treg cells, Tr1 cells secreting IL-10, CD8+ suppressor cells, natural killer T cells, CD4-CD8-T cells, and γδ T cells [45]. While CD4 regulatory T cells have been well characterized, the development, differentiation and activation of other T regulatory populations are still a matter of debate.

Overall, T reg cells may limit immune responses to self-antigens preventing autoimmunity as well as downregulating responses to foreign antigens (including allergens and food antigens), by a number of mechanisms that are being characterized, and that are used differently by different Treg subsets [45].

Basically, the main suppression mechanisms include:(1)Production of inhibitory cytokines such as IL-10, TGF-β and IL-35;(2)Direct cytotoxic activity via granzyme A/B and perforin;(3)Inhibition by cell–cell contact through co-inhibitory receptors including CTLA4, PDL1, LAG3, TIGIT, TIM3, NKG2A;(4)Metabolic perturbation of T effector cells by subtraction of IL-2, production of adenosine via CD73 and CD39 ATP-ectoenzymes, induction of IDO in dendritic cells (DCs) and others.

In CeD, studies investigating Treg cells have been mainly focused on Tr1 and Foxp3+ Treg cells, but recent reports have also investigated TCRγδ+ IELs and CD8+ T cells with regulatory activity, as reviewed in the next sections. 

### 3.1. Type 1 Regulatory T Cells

Tr1 cells are one of the first CD4+ Treg cell populations described, characterized by the high expression of IL-10. They are involved in the prevention of immune responses to both foreign and autoantigens, and particularly in the maintenance of long-term tolerance [22,46,47,48]. 

Tr1 cells develop in the periphery and IL-10, mainly produced by tolerogenic dendritic cells [IL-10 DCs], is considered their principal generation factor [47,48]. 

A distinct feature of Tr1 cells is that their suppressive activity is dependent on TCR activation. After the antigen-specific TCR-binding, Tr1 cells produce predominantly IL-10 and TGF- β but, depending on the microenvironment, they can also produce intermediate amounts of IFN-γ and IL-5, and little or no IL-2, IL-4 and IL-17, with a cytokine profile distinct from those of Th1, Th2 and Th0 subsets [47,48]. 

IL-22 has also been recently reported to be produced by Tr1 [49]. 

As other Tregs, Tr1 are anergic and proliferates poorly in response to antigen stimulation, but cytokines such as IL-2 and IL-15 are potent growth factors [46].

In addition to the peculiar cytokine profile, Tr1 cells are identified by surface markers as LAG3+CD49b+ memory CD4 T cells; they also display high expression of CCR5, CD2, CD18, CD226, and co-inhibitory receptors such as PD-1, TIM-3, CTLA-4, and PD-L1, while the transcription factor FoxP3 is not constitutively expressed [47,48].

Tr1 mainly perform their suppressive function trough cytokine secretion but it has been demonstrated that they may also act by other inhibitory mechanisms [47,48].

Gut mucosa is a rich source of both IL-10 and TGF-β and it has been demonstrated that antigen-specific Tr1 cells, are effective in the maintenance of tolerance to food antigens [50]. 

Gluten-specific Tr1 suppressive cells have been described both in intestinal mucosa of celiac patients [22] (Table 1) and in a transgenic mouse model of CeD [51]. In a pivotal study, we generated gliadin-specific short-term CD4+ T cell lines from duodenal biopsies of CeD patients, in presence or absence of IL-10 [22]. As a result, IL-10-generated TCLs [IL10-iTCLs] were less responsive to gliadin stimulation in terms of IFN-γ production and cell proliferation. Further, single T cell clones were isolated from IL-10-iTCLs and characterized for their cytokine profile, to identify possible Tr1 cells. Upon activation with gliadin or polyclonal stimuli, the majority of gliadin-responsive TCCs had a Th0 phenotype (secretion of IL-2, IL-4, IL-10 and IFN-γ, and high proliferative rate), but a percentage of TCCs with the typical Tr1 cytokine profile (production of IL-10 and IFN-γ, but little or no IL-2 or IL-4 and low proliferative rate), were also expanded. Importantly, these Tr1 cell clones suppressed proliferation of pathogenic Th0 cells, in co-cultures assays (Figure 1a). 

According to the development of gluten-specific Tr1 cells in the periphery in vivo, there are also the results of studies in transgenic mice expressing human HLA-DQ2.5 and a gliadin-specific, humanized, T-cell receptor [51,63]. In these studies, the authors found that ingestion of deamidated gliadin-induced expansion of gliadin-reactive T cells with a Tr1-like phenotype, mainly in the spleen, and not, as expected, in the mesenteric lymph nodes. Importantly, these gliadin-reactive T cells had regulatory functions, because transfer of the cells, suppressed a gliadin-induced, delayed-type hypersensitivity response. In addition, it was suggested a certain plasticity between Th0, Th1 and Tr1 gliadin-specific T cell clones [51]. Although gliadin behave differently from other food antigens in the exploited mouse model, these studies support that the physiological response to deamidated gliadin in vivo is a tolerogenic response, suggesting that Tr1 cells may have a protecting role, in particular in at-risk subjects carrying HLA-DQ2.5 molecules who do not develop CeD.

The presence of gluten-specific Tr1 cells in non-celiac HLA-DQ2.5-positive subjects has been investigated only in one study [58]. In this study, using a gluten-tetramer based approach, Christophersen et al. failed to isolate Tr1 cells among gluten-tetramer binding T cells from peripheral blood mononuclear cells (PBMCs) of both celiac and non-celiac, thus concluding that Tr1 cells are not involved in the protection of healthy subjects from CeD development. Nevertheless, additional studies are required to address possible differences in Tr1 cells in CeD and controls, in order to clarify several aspects such as the existence of Tr1 cells specific for native gliadin, or possible differences between HLA-DQ2 and other DQ haplotypes.

### 3.2. Foxp3+ Treg Cells

Another well-known CD4+ regulatory cell population, commonly referred as Treg, is characterized by high levels of IL-2 receptor α chain (CD25) and master transcription factor Foxp3 (CD4+CD25^high^Foxp3+ cells). This Treg family is further subdivided in two main subpopulations: thymus-derived Treg cells (tTreg, also called naturally occurring nTreg) that are early produced in the thymus and migrate into the peripheral blood to maintain tolerance toward self-antigen, and peripherally derived or induced Treg (pTreg or iTreg) that are generated in the peripheral lymphoid organs upon exposure of naïve CD4+CD25-Foxp3− T conventional (Tconv) cells to small doses of cognate antigens. Foxp3+ Treg cells can be also developed from naïve T cells in vitro, by TCR stimulation in presence of additional factors such as IL-2 and TGF-β [64]. 

In addition to TGF-β, playing a crucial role in the development of Foxp3+ Treg not only in vitro but also in vivo, another important Treg generation factor is retinoic acid, produced in the gut [65]. Since Foxp3 is an intracellular molecule that can be transiently expressed also by other T cells, additional markers, in particular surface molecules, are required to unequivocally identify Treg, in human. To this scope, the combination of CD4+CD25^high^CD127^low^ surface markers has been documented to be effective [66]. 

However, several other markers are being reported to discriminate Treg from Tconv and T effector cells (Teff), as well as to distinguish between tTreg and iTreg, and activated from inactivated Treg cells, in human [64,66]. 

Differently from Tr1, Foxp3+ Treg cells require antigen stimulation to expand in vivo but are able to perform their inhibitory tasks also in the absence of TCR stimulation [67,68,69]. Once activated, Foxp3+ Treg cells are able to suppress Teff cells using all the four mechanisms cited above [64]. 

In CeD, intestinal Foxp3 expression has been evaluated by RT-PCR in several works, and all of them reported an increased Foxp3 expression in patients with untreated CeD, compared to controls [18,52,53]. These results were confirmed also by immunohistochemistry [15,18,21,23,52,53,70] and by flow cytometry [15,18,23,53]. CeD patients at gluten-free diet (GFD), still showed an increased number of Foxp3+ cells, although lower if compared to active CeD [61] (Table 1).

Accordingly, it has been demonstrated that in CeD patients, the histological Marsh grade correlated with the mean number of Foxp3+ cells in the lamina propria and with transglutaminase type 2 serum levels [21]. 

These data indicate that the immune system is attempting to control the persistent inflammation either by recruitment of Treg cells from blood to tissue, or through the expansion of such regulatory T cells in the mucosa. 

Recently, it has been shown that the Foxp3+ population is increased in the oral mucosa of CeD subjects, concomitantly with mucosal damage, in both treated and untreated CeD, suggesting also in this anatomical site a recruitment of Tregs with a “repair” phenotype [62].

Importantly, even mucosa of potential CeD (subjects with normal mucosa histology but serological markers of CeD) shows increased numbers of Foxp3+ Tregs compared with normal mucosa, although lower than in active CeD [15,52]. Therefore, the low-grade inflammation in potential CeD patients could reflect active regulatory mechanisms, preventing progression toward mucosal damage.

Whereas data reported for duodenal mucosa are consistent with an upregulation of Foxp3, which correlates with the degree of inflammation, conflicting results have been reported from peripheral blood studies (Table 1). Some studies reported no differences in PBMCs Foxp3+ T cells from active CeD patients, compared to healthy subjects [54,57,58]. 

Conversely, Kumar et al. [60] showed a reduction in Foxp3+ Treg cells in both treated and untreated CeD patient’s PBMCs when compared to controls, with similar values between untreated and treated CeD [60]. In other studies [55,56,71], a higher expression of PBMCs Foxp3+ Treg cells was observed in untreated CeD patients compared to controls. 

Regardless of Treg cells frequency, some studies have reported that such cells may be impaired in their capacity to downregulate local Teff cell functions or, conversely, that Teff cells may fail to respond to Tregs. 

An interesting paper by Serena et al. [72], analyzed the expression of specific Foxp3 isoforms, comparing the full length (FL) to the alternatively spliced isoform D2, since the last isoform cannot properly downregulate the Th17-driven immune response. Intestinal biopsies from patients with active CeD showed increased expression of Foxp3 D2 isoform over FL, while both isoforms were expressed similarly in control subjects, thus suggesting a possible defect in the Foxp3+ Treg function in atrophic celiac mucosa. 

Suppressive ability of CD4+CD25+ Foxp3+ Treg cells isolated from PBMCs of CeD patients and controls has been investigated using autologous CD4+CD25- T responders (Tresp) cells, with conflicting results. Some studies reported a similar suppressive capability on proliferation of Tresp cells, of CD4+CD25+ Treg cells from peripheral blood of untreated, treated CeD and controls [23,55]. Conversely, other studies, reported that Foxp3+ Treg cells of untreated CeD, could not efficiently downregulate Tresp cell functions [18,60]. 

Interestingly, Hmida et al. [18] conducted similar experiments using as Tresp cells, not only CD25-PBMCs but also intestinal lymphocytes (CD25- LPLs and IELs) [18]. By co-cultures experiments the authors demonstrated that, CD25- PBMCs, CD25-LPLs and IELs from active celiac patients were partially or not inhibited by peripheral Treg cells, either autologous or heterologous. Conversely, peripheral Tregs from active CeD patients could efficiently downregulate proliferation and IFN-γ production of PBMCs, LPLs, and IELs from controls. Thus, the authors concluded that the functional defect was not in peripheral Tregs but in intestinal Tresp lymphocytes. These results were in accordance with a previous study, where it was shown that intestinal T lymphocytes become unresponsive to the immunoregulatory cytokine TGF-β, and that IL-15 overexpression was involved in the resistance mechanism [73].

In line with the data that IL-15 may interfere with immune regulation, we have demonstrated that in active CeD patients, IL-15 was able of making Tresp cells resistant to the regulatory effects of CD4+CD25+ Treg cells [23]. In particular, in this study CD4+CD25+ T cells were directly isolated from intestinal biopsies of patients, and their suppressive ability was evaluated on autologous CD4+CD25− Tresp cells. The data showed that intestinal Treg, as well as peripheral blood Treg from celiac patients, are not functionally deficient. Nevertheless, in active CeD patients, IL-15 impaired the functions of Treg cells making Tresp cells refractory to the regulatory effects of Treg cells, in terms of proliferation and production of IFN-γ (Figure 1b).

Although this phenomenon was nonspecific for CeD patients, this effect was less marked in controls than in CeD, and the greater sensitivity to IL-15 of CeD patients is likely to be due to their increased expression of IL-15 receptor [23]. It was also shown in potential CeD that Foxp3+ T cells were not impaired by IL-15, more likely due to the reduced expression of IL-15 receptor [15]. 

We have also shown that Foxp3+ cells were increased in response to gliadin, in ex vivo organ culture experiments, thus suggesting that a gliadin-dependent expansion of Foxp3 Treg may occur in vivo, in celiac patients [23]. 

This expansion has been later demonstrated in a study investigating the gluten-specific CD4+ T cells recirculating in the peripheral blood of CeD patients at GFD that underwent a short oral gluten challenge [59]. By exploiting the antigen-induced co-expression of CD25 and CD40, it was showed that most of peripheral blood cells activated in response to deamidated gliadin peptides, were Foxp3+ CD39+ Treg cells. Nevertheless, when these gluten-expanded Treg cells were used in functional assays, they were impaired in suppressive activity, compared to polyclonal Treg cells from the same subjects, indicating that Treg cell dysfunction might be a key contributor to disease pathogenesis. 

As for the gluten-specific Tr1 cells, gluten-specific CD4+CD25^high^CD127^low^ cells were also not found in HLA-DQ2.5-positive healthy subjects [58]. 

### 3.3. CD8 T Lymphocytes with Regulatory Activity

In addition to CD4 Treg cells, mainly resident in intestinal lamina propria, populations of immune cells can be found within the intestinal epithelial cell layer, such as IELs, which consist mostly of CD8+T cells. Though most of those IELs express T cell receptor (TCR)-alpha beta chains (αβ), CeD is characterized by an increase in TCRγδ+ IELs that remain elevated even after removal of gluten from the diet and whose functional significance in CeD is still under investigation [74]. It has been hypothesized that CD8+TCRγδ + may play a role in the preservation of intestinal homeostasis by regulating the mucosal immune response and/or by contributing to the epithelial cell layer maintenance. Functional properties of TCRγδ+ IELs can be mediated by NK receptors (such as NKGD) expressed on a fraction of these cells [75]. At the same time, CD8+ TCRαβ+ IELs have been suggested to kill intestinal epithelial cells (IECs) in an NKG2D-MICA-dependent manner during CeD [76]. TCRγδ+ IELs might be activated by the same stimulus to induce protection, but more evidence is needed to determine whether they are protective or pathogenic during CeD. These IELs are able to produce KGF, and it has been demonstrated that they promote IEC turnover, and have a protective role in barrier disruption models of colitis [77]. In addition to KGF, TCRγδ+ IELs also secrete TGF-β, which contribute both to epithelial damage healing and to immune-regulation [78]. In line with these findings, it has been demonstrated that a fraction of celiac patient’s TCRγδ+ IELs expressing the inhibitory receptor NKG2A responds to the ligation of such receptor, with a rise in TGF-β production [44]. Importantly, these cells downregulate the cytotoxic function of TCRαβ+ IELs by decreasing the expression of molecules such as granzyme-B, IFN-γ and NKG2D, through a mechanism partially mediated by TGF-β.

In addition to TCRγδ+ CD8+ cells, a regulatory activity has been recently proposed also for a subpopulation of TCRαβ+ CD8+ cells. A subset of regulatory CD8+ T cells that express Ly49 has been described in mice [79,80], able to reduce autoimmunity in a model of experimental autoimmune encephalomyelitis [81] but also with documented functions in several other disease settings [79]. These cells are Foxp3- but TGF-β is necessary to maintain their regulatory identity. Mechanisms of action involve IL-10 secretion, killing via granzyme/perforin, and induction of apoptosis by FAS/FASL interaction. Ly49 receptors are a family of NK receptors, including some inhibitory ones. Their functional analogues in humans are killer-cell immunoglobulin-like receptor (KIR) genes. Recently, Li et al. [82] reported a regulatory CD8+ T cell subset expressing KIRs homologs of inhibitory Ly49F (KIRDL1/2/3) and carrying an expression profile similar to mouse Ly49+CD8+ T cells, in human. Interestingly, they found that KIR+CD8+ T cells were more abundant in blood and intestinal mucosa of patients with CeD, compared to healthy subjects. More strikingly, KIR+CD8+ T cells were able to block the expansion of gluten-specific CD4+ T cells in vitro. The authors provide also preliminary evidences on suppression mechanisms that seem to be antigen specific, cell-contact dependent, mediated by apoptosis of CD4+ T cells and not linked to IL-2 consumption (Figure 1d). Nevertheless, the mechanisms that drive the differentiation, expansion, and activation of KIR+CD8+ T cells in celiac patients and, in general, in autoimmunity in humans are unknowns, and several issues are opened [82,83].

## 4. Regulatory T Cells in the Context of CeD Pathogenesis

As reviewed in other excellent comprehensive articles [1,2,6], immune-pathogenesis of CeD is a complex mosaic, where different factors are needed to interplay for promoting the intestinal damage. Briefly, gluten is not completely digested by gastro-intestinal enzymes, and peptides that manage to cross the epithelial barrier are subjected to deamidation by tTG2, into the lamina propria. Here, peptides are taken up by DCs and presented to CD4+ T cells, in the context of HLA-DQ2 or DQ8 molecules, thus promoting the differentiation of naïve gluten-specific CD4+ T cells into Th1 effector T cells, in induction sites. In addition to production of inflammatory factors (IFN-γ, IL-21, IL-2), Th1 provide help to gluten-specific and TG2-specific B cells, secreting anti-gliadin and anti-TG2 antibodies [1,2]. At the same time, cytotoxic intraepithelial cells (IE-CTLs) receive activation signals from CD4+ Th1 cells, but also from the epithelial cells, expressing stress molecules and high amounts of IL-15. Overall, CD8+ IELs expressing activating NK receptors are able to kill stressed epithelial cells expressing ligands for such receptors, through perforin and granzymes-mediated mechanisms [1,2]. Further, cytotoxic T lymphocytes can be directly activated by gliadin peptides into the lamina propria, and induce the apoptosis of ECs presenting such gluten peptides on HLA-class I molecules [84]. 

In this context, when gluten is ingested, together with expansion of pro-inflammatory lymphocytes, regulatory T cells producing large amount of IL-10 are also increased [7,12,13,14,15]. As discussed in the previous sections, at least four different T cells with suppressive activity are present in the intestinal mucosa and/or in the peripheral blood of CeD patients, which could be activated directly by gluten, or indirectly by microenvironmental signals. Such regulatory subsets may expand locally, or may be recruited from the periphery to the inflamed tissue. In spite of their origin and activation, suppressive cells may act directly or indirectly on the main orchestrators of the mucosal inflammation, including CD4+ T cells, cytotoxic IELs and DCs (Figure 1). Notwithstanding, mechanisms put in place by Treg cells are not sufficient to keep under control the inflammation triggered by gluten, in active CeD mucosa. Strikingly, in potential CeD, where the ratio IL-10/IFN-γ is higher compared to active CeD, the inflammation is well controlled [15]. A possible factor explaining the reason of such differences, could be the excessive production of IL-15 associated with increased levels of IL-15R on lymphocytes, interfering with the suppressive capacity of Treg cells [23]. Moreover, the role of additional elements such as virus infection and microbiota alterations in immunoregulatory pathways is still a matter of debate. 

## 5. Therapeutic Applications of Treg Cells in CeD

The pathogenesis of CeD is triggered by the loss of tolerance towards gluten peptides. 

On the other hand, as discussed above, several studies have suggested that the suppressive effect of Treg cells might be impaired in vivo in CeD by the inflammatory microenvironment, and their dysregulated function may contribute to sustain and expand the local inflammatory response. 

Studies based on increasing the frequency or reinforcing inhibitory function of Treg cells, with the aim to restore immune tolerance at the inflamed tissue, have shown interesting results in several inflammatory disorders such as multiple sclerosis (MS), type 1 diabetes (T1D) and systemic lupus erythematosus (SLE) [85,86,87], thus encouraging similar approaches for CeD treatment. 

Among these immunomodulating therapies, there are:-Approaches based on in vivo administration of drugs such as rapamycin, or biologicals, such as IL-10 or low-dose IL-2, to suppress Teff cells and promote Tregs.-Approaches resting on the administration of the autoantigen by using lentiviral vectors or antigen-specific nanoparticles, promoting tolerogenic cells and the expansion of Treg cells.

Moreover, cell-based therapies have been developed to enhance Treg cell specificity, function and number. This can be achieved by expanding Tr1 cells expressing IL-10 [88], Tregs expressing a natural repertoire of polyclonal TCRs [89] or Tregs that have been ex vivo-engineered to express a specific autoantigen receptor, such as a TCR, or a chimeric antigen receptor (CAR) [90,91]. 

DCs also offer a cell-based therapeutic way to restore tolerance and prevent autoimmunity [92]. 

Tregs can generate a tolerogenic phenotype in DCs, which can contribute to the rescue of immune tolerance. Tolerogenic DCs (TolDC), generated ex vivo, could be administrated, in vivo, to suppress autoimmunity in diseases such as T1D [93,94]. 

Other approaches have used mature DCs to expand antigen-specific Tregs [95], as for the generation of tolDC by genetic engineering of monocytes (CD14+) with lentiviral vectors co-encoding for immunodominant antigen-derived peptides and IL-10 [96]. 

Therefore, therapeutic approaches aiming to restore tolerance to gluten, and/or to correct Treg functioning, would be a great step forward to protect CeD patients from excessive immune response, to reinstate intestinal homeostasis and, possibly, to allow improvement in patients outcomes.

In this regard, some studies have investigated the possibility of restoring immune tolerance to gluten or targeting the gluten-induced immune activation, as promising therapeutic options to GFD for CeD patients.

More recently, Passeri et al. [96] demonstrated that human DC^IL10/glia^, generated by genetic engineering of monocytes with lentiviral vectors co-encoding for immunodominant gliadin-derived peptides and IL-10, inhibit pathogenic gliadin-specific CD4+ T cells and promote the differentiation of gliadin-specific Tr1 cells (Figure 2a) in PBMCs from HLA-DQ2+ CeD patients [96].

Other approaches have been developed using in vivo antigen-delivery to induce a tolerogenic inhibition of specific immune response [97,98]. 

In particular, several evidences demonstrated that nanoparticles could be useful to deliver specific antigens that induce tolerogenic inhibition via a noninflammatory process. These nanoparticles interact with a specific receptor on APCs, named MARCO (macrophage receptor with collagenous structure), leading to a tolerogenic presentation of Ag-to-Ag-specific T cells [99]. 

This tolerance strategy leads to anergy within Ag-specific Teff cells and activate populations of Ag-specific regulatory T cells [97,98,99,100,101,102,103]. 

In this context, gliadin encapsulated in nanoparticles, namely TAK-101 (formerly TIMP-GLIA, Tolerogenic Immune Modifying nanoparticles) (Figure 2b), is under clinical development. Firstly, this approach has been shown to be effective in a mouse model of CeD [104,105], where intravenous infusion of gliadin-encapsulating nanoparticles inhibited the proliferation, and the IFN-γ and IL-17 secretion, of gliadin-specific T cells, while increasing the frequency of FoxP3+ Treg cells. 

Subsequently, a clinical trial performed [106] through intravenous administration of TAK-101 nanoparticles in CeD patients on a GFD underwent to gluten challenge, showed that Ag-specific T cell response induced by the gluten challenge was reduced compared to a placebo group, indicating that TAK-101 acts in an Ag-specific manner. Nevertheless, the injection of nanoparticles did not lead to an increase in peripheric Tregs. However, it may be possible that antigen-specific Treg cells localize at the site of antigen presentation, without being found in the circulation. 

A possible cellular therapy representing a great step forward in the treatment and, possibly, the cure of chronic inflammatory conditions, is based on the use of stem cells. Both hematopoietic stem cells (HSCs) and mesenchymal stem/stromal cells (MSCs) have been employed in the treatment of refractory cases with promising results. By virtue of the lack of immunogenicity and of the ability to favor tissue regeneration and expansion of T cells with regulatory function, MSCs seem the best candidate for clinical application. MSC are multipotent non-hematopoietic cells present in different tissues, including bone marrow, amniotic fluid, umbilical cord, and placenta, which due to their immunomodulatory characteristics are considered as new therapeutic agents in the cell-based therapy of autoimmune and immune-mediated diseases [107,108,109,110,111,112,113,114]. 

The beneficial effects resulting from the transfer of factors and molecules belonging to the cell secretome, released by MSCs, could be reproduced to a limited extent [115].

MSCs may be useful in view of their varied immunomodulatory properties such as their ability to promote Treg development. A recently published study demonstrated the important role of MSC for the in vitro expansion of the Treg population from purified human conventional CD4+ T cells. The resulting cells closely resembles natural Treg in terms of phenotype and suppressive ability [116]. These data suggested a potential application of these MCS-induced Treg as therapy in disease where the immune tolerance is broken.

Previously, Ciccocioppo et al. showed an increased percentage of circulating and mucosal Tregs in Crohn’s disease (CD) patients after MSC therapy [117]. Subsequently, the same group [118], provides the first evidence on the ability of bone marrow-derived MSCs to affect gliadin-specific T-cell reactivity. 

Particularly, a reduction in CD4^+^ T cells, concomitantly to an increase in FoxP3+ T reg cells, was found in T-cell lines cultured with MSCs. Moreover, a significant reduction in expression of interleukin IL-21, IFN-γ and IL-10 together to an upregulation of TGF-β, IL-6 and IL-8 (Figure 2c) was detected [118]. 

Such results make MSCs an attractive tool as a new therapeutic approach in CeD. 

The use of MSC as adjuvant therapy combined with conventional pharmacological approaches has also been considered. As previously discussed, in active CeD, IL-15 may interfere with immune regulation [23,24], therefore, treatments to improve the Treg population based on MSC therapies or the administration of cytokines known to promote their survival and expansion, as IL-10/TGF-β or inhibition of pro-inflammatory cytokines known to promote Treg dysfunction/cell death, such as anti-IL-15, could be also explored in CeD.

## 6. Conclusions

Celiac intestinal mucosa harbors two subsets of CD4+ Treg cells, Tr1 and Foxp3+ T cells. Many factors, such as IL-15, largely expressed in the CeD mucosa, may interfere with the function of Treg cells, thus contributing to the loss of intestinal homeostasis and promoting chronic inflammation.

On the other hand, novel T cell subsets with regulatory activity are emerging, and their further characterization is of great interest. As Treg cells exist naturally in the human gut mucosa and maintain intestinal homeostasis, using methods to enhance their numbers and/or function is a possibility worthy of pursuit as a new therapeutical approach to re-establish tolerance to gluten in patients with CeD. Treg immunotherapies based on infusion of autologous TolDC or MSCs, or on enhancement of Treg numbers and function via administration of nanoparticles, remain possible strategies to be implemented in CeD.

## Figures and Tables

**Figure 1 ijms-24-14434-f001:**
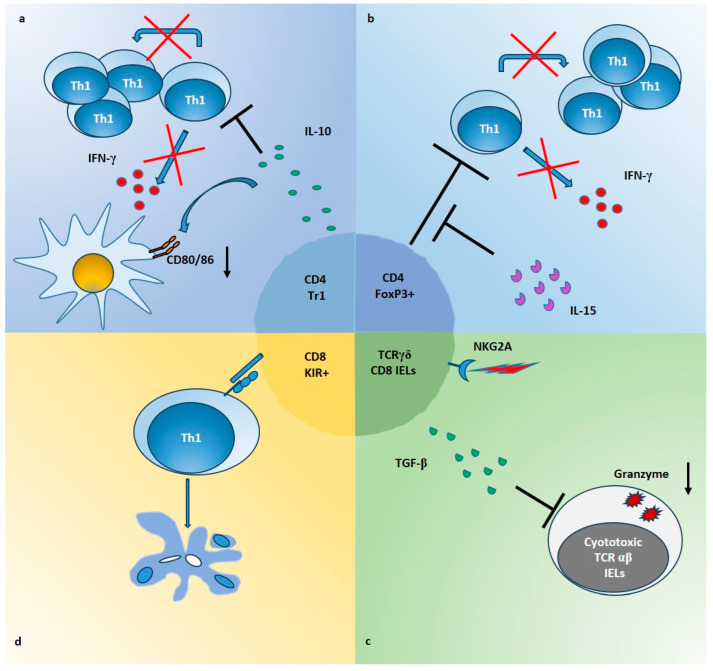
Mechanisms of Treg suppression in CeD intestinal mucosa. (**a**) CD4+ Tr1 cells producing high amounts of IL-10, inhibit the proliferation and the IFN-γ production of gliadin-responders Th1 cells (red crosses and black lines). In addition, IL-10 downregulates the expression of CD80/CD86 on APCs (black arrow), induced by gliadin stimulation in intestinal mucosa. (**b**) Foxp3+ Treg cells, which are expanded in CeD mucosa, are able to suppress proliferation and IFN-γ production of CD4 Tresp cells (red crosses and black lines) but IL-15 impairs their function in intestinal mucosa (black lines). (**c**) TCRγδ+ CD8+ IELs infiltrating the epithelium of CeD gut, increase the production of TGF-β following the ligation of NKG2A. These cells are able to inhibit expression of molecules such as granzyme B (black arrow), in cytotoxic TCRαβ+ IELs, through a mechanism partially dependent on TGF-β. (**d**) KIR+CD8+ T cells can block the proliferation of gliadin-specific Th1 cells by induction of apoptosis (color circles indicate apoptotic bodies), through a cell-cell contact mechanism not yet defined (blue molecules).

**Figure 2 ijms-24-14434-f002:**
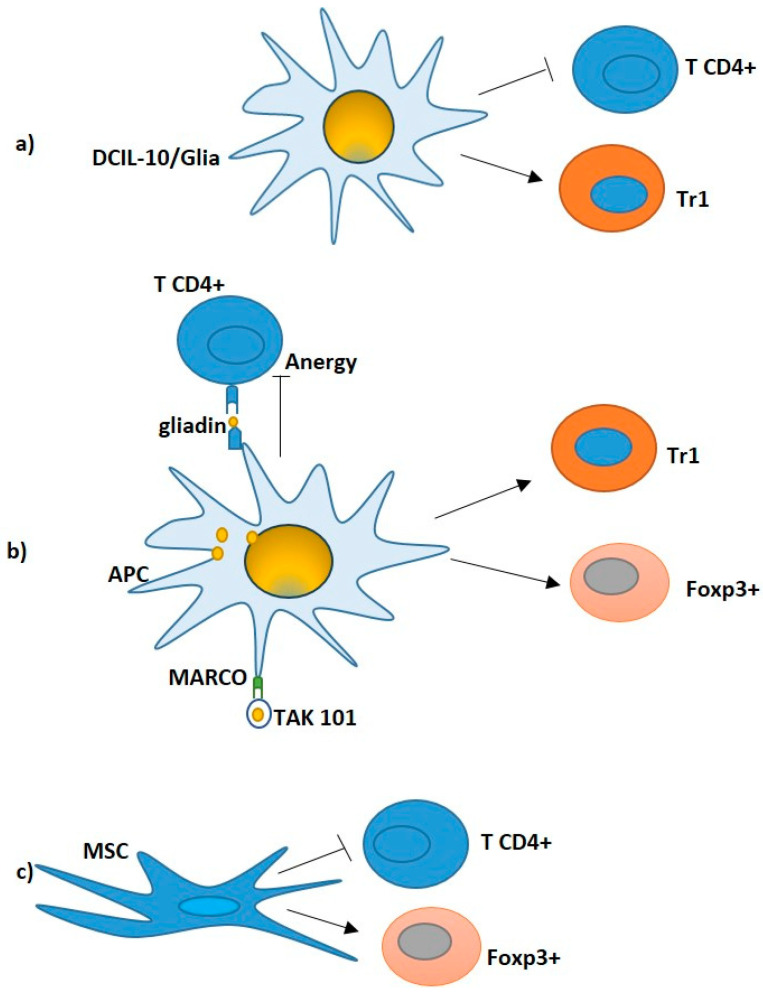
Potential Treg- cell based therapy in CeD. (**a**) DCIL-10/Ag modulate pathogenic gliadin-specific T cells, showing efficient inhibition of pathogenic CD4+ T cell functions (black line) and induction of gliadin-specific Tr1 in vitro (black arrow). (**b**) TAK101: Nanoparticles encapsulating gliadin interacted with tolerogenic APCs, leading to the release of TGF-β and IL-10. Gliadin T cell epitopes are processed and presented to specific T cells. The activation of both FOXP3+ Tregs and IL10-producing Tr1 cells (black arrows) supports the activation of tolerance. (**c**) MSC: T-cell lines cultured with MSCs led to a reduction in the CD4+T cells (black line) and expansion of the FoxP3+ T regulatory cells (black arrow), in which a significant decrease in interleukin IL-21, IFN-γ and IL-10 paralleled by an upregulation of transforming growth factor-β1, IL-6 and IL-8 were observed.

**Table 1 ijms-24-14434-t001:** CD4+ T regulatory cells in CeD patients.

Treg Cell Phenotype	Patients/ Specimens	Main Findings	Ref.
Tr1 cells (IL10^hi^, IFN-γ^pos^, IL4^low/neg^, IL-2^low/neg^ )	Untreated and treated adults/ Duodenal mucosa	Suppression of gliadin-specific Th0/Th1 response by Tr1 cells.	[22]
CD25+Foxp3+	Children with Overt or potential CeD/ Duodenal mucosa	Increase in Foxp3 mRNA, CD25+ cells, and FoxP3+ cells.	[52]
CD25+Foxp3+	CeD children with or without T1D/ Duodenal mucosa	Increase in Foxp3 mRNA, CD25+ cells, and Foxp3+ cells in partial or subtotal villus atrophy.	[53]
CD4+CD25+Foxp3+	Children/ Peripheral Blood	Impaired suppression activity of Treg.	[54]
CD4+CD25+Foxp3+	Untreated and treated adults/ Peripheral blood	Increase in CD4+CD25+Foxp3+ frequency in untreated.	[55]
CD4+CD25+Foxp3+	Untreated and treated Adults/ Duodenal mucosa and peripheral blood	Increase in CD4+CD25+Foxp3+ in untreated. Gliadin-dependent expansion of Foxp3 Treg. Inhibition of Treg suppression activity by IL-15.	[23]
CD4+CD25+Foxp3+	Untreated and treated adults/ Duodenal mucosa and peripheral blood	Increase in FoxP3 mRNA and CD4+CD25+Foxp3+ LPLs in biopsies of untreated. Resistance of LPLs and IELs to the suppressive activity of peripheral blood Tregs.	[18]
CD4+CD25+Foxp3+	Children with overt or potential CeD/ Duodenal mucosa and peripheral blood	Increase in mucosal Foxp3+CD25+CD4+ cells. No influence of IL-15 on intestinal Tregs from potCeD.	[15]
CD62L+Foxp3+ nTreg and CD62-CD38+Foxp3+ iTreg	CeD children and treated or refractory adults/ Duodenal mucosa and peripheral blood	Increase in circulating nTreg in adult treated CeD and RCeD. Increase in Foxp3 + cells in LPLs of children and adult CeD.	[56]
CD4+CD25+Foxp3^high^CD127^low^	Children with or without T1D/ Peripheral blood	No differences.	[57]
CD4+Foxp3+CD25+ T cells and IL-10^high^CD4 Tr1 cells	HLA-DQ2.5+ healthy and CeD subjects/ Duodenal mucosa and peripheral blood	No detection of Foxp3+ and of Tr1 cells within gluten-tetramer binding CD4+ T cells.	[58]
CD39+Foxp3+ CD4+ T cells	Treated adults undergone to SGC/ Peripheral blood	Increase in circulating CD39+Foxp3+ CD4+ Treg cells after gluten challenge. Impaired suppressive function of gluten-specific Treg cells.	[59]
CD4+Foxp3+CD25+ nTreg and CD4+Foxp3+ iTreg	Untreated and treated Children/ Peripheral blood	Decrease in nTreg and iTreg frequency in both untreated and treated.	[60]
CD4+CD25+Foxp3+	Treated adults/ Duodenal mucosa and peripheral blood	Increase in CD25 and Foxp3 mRNA expression in duodenal mucosa.	[61]
CD4+Foxp3+	Untreated and treated Adults/ Oral mucosa	Increase in Foxp3+ cells.	[62]

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
