# Peer review of "Role of Regulatory T Cells and Their Potential Therapeutic Applications in Celiac Disease"

_ijms, 2023, doi:10.3390/ijms241914434_

Round 1

Reviewer 1 Report

In this review, Camarca and Colleagues addressed the role of Treg cells and suppressive cytokines in controlling gluten-dependent inflammation and autoimmunity in the celiac intestinal mucosa. They also discussed the potential of Treg cells for the development of novel emerging therapeutic strategies.

The manuscript is well-written and presented. However, since the Treg cell population is a well-characterized immune cell population involved in other autoimmune diseases and autoantibody development, in my opinion, in this review the Authors should include a section to discuss the autoantibody repertoire developing in celiac patients. Discussing the connection between celiac disease and Type 1 Regulatory T cells and pathogenic mechanisms, the authors could further address the potential role of immunological mechanisms as suggested by the reported positivity of serum autoantibodies recognizing different autoantigens. 

In particular, other than anti-tissue transglutaminase and deamidated gliadin peptides, previous studies have described other autoantibodies with clinical significance in celiac disease patients. 

For instance, the association between severe villous atrophy and anti-actin IgA autoantibodies which disappear after a gluten-free diet and mucosal recovery, as previously demonstrated (Anti-actin IgA antibodies in severe coeliac disease. Clin Exp Immunol. 2004 Aug;137(2):386-92. doi: 10.1111/j.1365-2249.2004.02541.x.). Likewise, the demonstration of anti-neuronal antigens and anti-gangliosides autoantibodies in coeliac patients with neurological manifestations, as previously demonstrated (Sera of patients with celiac disease and neurologic disorders evoke a mitochondrial-dependent apoptosis in vitro. Gastroenterology. 2007;133:195-206; Anti-ganglioside antibodies and celiac disease. Allergy Asthma Clin Immunol. 2021 May 28;17(1):53. doi: 10.1186/s13223-021-00557-y. ). Recalling this literature data would further highlight the crucial pathogenetic role of loss of immunological tolerance in coeliac disease.

-In this regard, the association between celiac disease and extraintestinal autoimmunity could also be explained, with the theory of "secondary autoimmunity". It has been supposed that, in celiac disease, an increased transglutaminase activity in inflammation, might generate additional antigenic neoepitopes by cross-linking or deamidating external viral, bacterial, nutritional, or endogenous functional/structural proteins as previously suggested (PMID: 11926568; DOI: 10.1016/s1590-8658(02)80053-6).

-Discussing the role of Anti-inflammatory cytokines in CeD, would be of clinical significance recalling the demonstration that celiac disease patients, before starting the gluten-free diet, very frequently develop antibodies to Saccharomyces cerevisiae, as a result of mucosal damage and increased mucosal permeability, as previously demonstrated (Anti-Saccharomyces cerevisiae and perinuclear anti-neutrophil cytoplasmic antibodies in coeliac disease before and after gluten-free diet. Aliment Pharmacol Ther. 2005;21:881-7.)

Author Response

We appreciated the suggestions of Reviewer#1 and we have modified the text by highlighting the importance of auto-antibodies in CeD diagnosis and pathogenesis in the introduction (lines 30-35), citing the suggested articles.

Nevertheless, since the CeD pathogenetic mechanisms have only been briefly overviewed in this review, we prefer avoiding to go through the details of the origin and consequence of these antibodies, just to maintain a balance between the review sections.

Reviewer 2 Report

The manuscript is a good review of the subject, it could be an excellent review after correction of the order of the references. In the references section some of the cites in the text are missing and several of them have two numbers, some of them are not complete, and so on. Also, there are cites in the text without the full reference in the last list.

In addition, the manuscript sounds so good as a hypothesis. I would like to see a short and strong section of theoretical proof of the hypothesis before the last section on therapeutic applications.

No comments

Author Response

Comment#1:  The manuscript is a good review of the subject, it could be an excellent review after correction of the order of the references. In the references section some of the cites in the text are missing and several of them have two numbers, some of them are not complete, and so on. Also, there are cites in the text without the full reference in the last list.

Authors replay: We apologize with reviewers for the mistakes in the reference order, more likely due to a problem during file uploading and conversion. We have corrected the references and added some more articles as suggested by reviewer 1.

Comment #2:

In addition, the manuscript sounds so good as a hypothesis. I would like to see a short and strong section of theoretical proof of the hypothesis before the last section on therapeutic applications.

Authors replay:

We thanks to the suggestions of Reviewer#2 and we have added in the manuscript a paragraph entitled “Regulatory T cells in the context of CeD pathogenesis”, lines 468-502. 

Round 2

Reviewer 1 Report

The Authors satisfactorily addressed the raised points and the manuscript can be accepted.

Author Response

We thanks the reviewer for the comments that assisted us in improving the quality of our review.

Reviewer 2 Report

It is unacceptable, the list of references is still wrong. Please follow carefully the guidelines for authors and published articles in the same journal to check the right way to write the section of References.

No problem

Author Response

We thanks the reviewer for its comments.

We have revised references and uploaded an edited manuscript version, following the reviewer recommendations.